# Mitoprotective Clinical Strategies in Type 2 Diabetes and Fanconi Anemia Patients: Suggestions for Clinical Management of Mitochondrial Dysfunction

**DOI:** 10.3390/antiox9010082

**Published:** 2020-01-18

**Authors:** Giovanni Pagano, Federico V. Pallardó, Beatriz Porto, Maria Rosa Fittipaldi, Alex Lyakhovich, Marco Trifuoggi

**Affiliations:** 1Department of Chemical Sciences, Federico II Naples University, I-80126 Naples, Italy; marco.trifuoggi@unina.it; 2Department of Physiology, Faculty of Medicine and Dentistry, University of Valencia-INCLIVA, CIBERER, E-46010 Valencia, Spain; federico.v.pallardo@uv.es; 3Institute of Biomedical Sciences, ICBAS, University of Porto, 4099-030 Porto, Portugal; bporto@icbas.up.pt; 4Internal Medicine Unit, San Francesco d’Assisi Hospital, I-84020 Oliveto Citra (SA), Italy; m.fittipaldi@asl.salerno.it; 5Vall d’Hebron Institut de Recerca, E-08035 Barcelona, Spain; lyakhovich@gmail.com; 6Institute of Molecular Biology and Biophysics of the “Federal Research Center of Fundamental and Translational Medicine”, 630117 Novosibirsk, Russia

**Keywords:** type 2 diabetes, Fanconi anemia, oxidative stress, mitochondrial dysfunction, mitochondrial nutrients

## Abstract

Oxidative stress (OS) and mitochondrial dysfunction (MDF) occur in a number of disorders, and several clinical studies have attempted to counteract OS and MDF by providing adjuvant treatments against disease progression. The present review is aimed at focusing on two apparently distant diseases, namely type 2 diabetes (T2D) and a rare genetic disease, Fanconi anemia (FA). The pathogenetic links between T2D and FA include the high T2D prevalence among FA patients and the recognized evidence for OS and MDF in both disorders. This latter phenotypic/pathogenetic feature—namely MDF—may be regarded as a mechanistic ground both accounting for the clinical outcomes in both diseases, and as a premise to clinical studies aimed at counteracting MDF. In the case for T2D, the working hypothesis is raised of evaluating any in vivo decrease of mitochondrial cofactors, or mitochondrial nutrients (MNs) such as α-lipoic acid, coenzyme Q10, and l-carnitine, with possibly combined MN-based treatments. As for FA, the established knowledge of MDF, as yet only obtained from in vitro or molecular studies, prompts the requirement to ascertain in vivo MDF, and to design clinical studies aimed at utilizing MNs toward mitigating or delaying FA’s clinical progression. Altogether, this paper may contribute to building hypotheses for clinical studies in a number of OS/MDF-related diseases.

## 1. Introduction

Redox and mitochondrial anomalies are recognized—since the pioneering report by Luft (1994) [1] on “mitochondrial medicine”—in an extensive number of disorders that are afferent to several medical disciplines [2,3,4,5], including diabetes, aging, and genetic, neurologic and neuropsychiatric, and cardio-vascular diseases. Thus, several clinical studies have attempted, with varying success, to provide adjuvant treatments aimed at counteracting OS/MDF-related disorders by means of essential mitochondrial cofactors [6,7]. As stated by Wesselink et al. [6], mitochondria-focused treatments can provide basic support in contrasting disease progression in a number of OS/MDF-related disorders. The recognized key molecules in mitochondrial functions are α-lipoic acid (ALA), coenzyme Q10 (coQ10), and carnitine (CARN), also termed mitochondrial nutrients (MNs) [8]. These are involved in three basic mitochondrial functions, namely the Krebs cycle (ALA), the electron transport chain (coQ10), and acyl transfer (CARN), as recognized since the early report by Palade in 1964 [9].

Mitochondria’s essential tasks rely on oxidative metabolism of nutrients in producing high-energy molecules such as ATP, thus reactive oxygen species (ROS) are a necessary by-product of these biotransformations, causing an intramitochondrial pro-oxidant state. Hence, mitochondria are commonly regarded as the main cell’s “power plant” and the first victim of this pro-oxidant state. As a lucky evolutive solution for mitochondrial—and cellular—survival, MNs both exert their cofactor roles in energetic—pro-oxidant—metabolism and behave as strong antioxidants [10,11,12].

Among the extensive number of OS/MDF-related disorders [2], the present review is focused on two apparently unrelated diseases, such as a pandemic disease as T2D and a rare genetic disease as FA. These may be seen as two paradigms both offering mechanistic insights of the two disorders and providing working hypotheses toward adjuvant treatments of these and other OS/MDF-related disorders.

## 2. Type 2 Diabetes

The involvement of OS in T2D pathogenesis has been demonstrated in the last 30 years [13,14,15,16,17]. The mechanistic roles of OS and MDF in T2D pathogenesis both include the primitive damage to pancreatic beta cells [18,19] and the major clinical consequences of T2D progression, such as micro- and macro-angiopathies, and retinal damage [20,21].

As a concurrent feature of T2D-associated prooxidant state, defective respiration and oxidative phosphorylation was reported since early studies of diabetic patients and in animal studies. At the purpose to counteract T2D-associated OS, the use of several antioxidants was suggested [22,23,24,25,26,27,28,29,30]. Thereafter, and up to recent studies, MDF was further investigated as a salient feature of T2D, while a mechanistic role of the antidiabetic drug metformin, and of an herbal antidiabetic formulation, was associated with improvement of mitochondrial function [31,32,33].

In view of the recognized roles of MNs in mitochondrial functions, a body of literature has focused on the beneficial outcomes of treating diabetic patients with each MN, leading to clinical improvements following patients’ treatment with ALA, or coQ10, or CARN, as summarized in Table 1 (reviewed in [7]). Since early clinical studies in the 1990s, ALA has been demonstrated to provide adjuvant support in ameliorating the clinical conditions of T2D patients [34,35,36,37,38,39,40,41,42,43,44,45,46,47,48], with the recognition of ALA as a prescription drug in diabetic patients by the German Drug Index [49].

Other clinical studies have investigated the adjuvant effects of either coQ10 [50,51,52,53,54,55] or CARN (or acyl-CARN) [56,57,58,59,60,61,62,63,64,65] in T2D patients, by improving blood pressure and glycemic control.

Unlike the above clinical trials, only utilizing one MN, a pilot study by Palacka et al. in 2010 [66] reported on the adjuvant treatment of T2D patients by administering limited doses of ALA (100 mg/day) and of coQ10 (60 mg/day). To the best of our knowledge, no other clinical studies focused on two-MN administration in T2D patients, and no report was found reporting on triple MN administration. Nevertheless, it should be mentioned that MN combinations were reported in either in vitro or in animal studies aimed at counteracting MDF and OS in experimental diabetes or in other MDF-related disorders [67,68,69,70,71,72,73]. This was the case, e.g., for the studies by Shen et al. [69,70] reporting on combined ALA and CARN mixture-induced protective effects in rat or murine cells; MN mixtures were also found to protect diabetic rats from immuno- and liver-dysfunction [72,73].

Based on the available knowledge of safe and successful use of each MN in T2D, and in several other disorders, one may suggest that safe administration of triple MN combinations might provide successful outcomes in counteracting T2D-related symptoms and progression. This working hypothesis warrants design of ad hoc experimental and clinical studies.

## 3. Fanconi Anemia

According to an extensive consensus in the scientific community, FA was defined by Grover Bagby in 2018 [74] as a «rare inherited syndrome characterized by progressive bone marrow failure (BMF) and a high relative risk of hematopoietic and epithelial malignancies in children who sometimes have characteristic developmental abnormalities, including short stature and radial ray defects. Since the first clinical report of this disease 90 years ago, important clinical advances have included the development of the gold-standard diagnostic test (chromosomal breakage responses when lymphocytes or fibroblasts are exposed to low doses of either mitomycin C or diepoxybutane).

Beyond the “canonical” definition linking FA to defective DNA repair [74], three major features of FA phenotype are herein discussed, both in the attempt to understand FA pathogenesis and to develop improvements in the clinical management of FA patients. These phenotypic features include excess T2D prevalence in FA patients, the in vitro/in vivo evidence for OS, and the occurrence of MDF, though so far confined to in vitro and molecular investigations.

### 3.1. Excess T2D Prevalence

An excess T2D prevalence is a well-established phenotypic feature in FA patients [75,76,77], as assessed by the International Fanconi Anemia Registry [75], along with excess prevalence (81%) of multiple endocrine abnormalities. This finding was confirmed by further studies [76,77], both in FA patients and in FA mice [78]. These links between FA and T2D are consistent with shared pathogenetic mechanisms, first including the role(s) of OS both in T2D and in FA. It is worth noting that excess T2D prevalence is not confined to FA, as far as a number of other genetic diseases [79,80,81,82,83] display excess T2D prevalence (along with OS features), independently of their multiple and different genetic deficiencies. Thus, the co-occurrence of T2D in these disorders suggests common deficiency(ies) focused on OS/MDF abnormalities.

### 3.2. FA: In Vitro and In Vivo OS

The occurrence of redox abnormalities in FA is recognized since early in vitro studies by Nordenson [84] and by Joenje et al. [85]. Subsequently, an extensive body of literature has assessed evidence for the pathogenetic roles of OS in FA gene products [86,87,88,89,90,91,92,93,94], gene transcripts [95], cultured cells [96,97,98,99,100,101,102,103], and in freshly drawn blood cells, plasma and urine from FA patients [104,105,106]. In brief, OS, as a “non-canonical” feature of FA [74], can be reasonably considered as a major pathogenetic feature of this disorder.

### 3.3. FA: In Vitro and Molecular Evidence for MDF

After the pioneering report by Mukhopadhyay et al. in 2006 [107], it became clear that a FA gene product (FA-G) was associated with MDF, opening the way toward the assumption that FA proteins may be related to the control of mitochondrial activity [108,109,110]. In recent years, a growing body of evidence has been provided assessing that other FA gene products, such as *FANCA, FANCC,* and *FANCD2* are connected to mitochondrial function [111,112,113,114] such as energetic function, altogether providing a link between OS and MDF in FA as well as in a number of cancer-prone genetic diseases, as discussed by Perrone et al. [115].

An involvement of MN in FA was notably suggested by Ponte et al. [102] who reported that L-CARN elicited a significant protective effect against DEB-induced OS, which was potentiated by ALA. Additionally, the same authors reported that ALA, in combination with *N*-acetyl-cysteine, improved the genetic stability in FA lymphocytes in vitro, significantly decreasing the spontaneous and DEB-induced chromosome instability associated with the cellular FA phenotype [103]. Thus, ALA-related function in MDF prevention may be consistent with a delayed disease progression.

One may conclude that the present state-of-art shows a persuasive link between OS and MDF in FA’s cellular and molecular phenotype. Thus, one may raise an imperative question about the in vivo relevance of these combined deficiencies, so far confined to molecular and in vitro investigations. An in vivo confirmation of the previously reported molecular and cellular MDF in FA is badly needed and very likely. Should a confirmation be provided, this would open gateways to innovative, mitochondrial-based clinical management for FA patients.

## 4. Safety and Toxicity of Mitochondrial Nutrients

### 4.1. ALA

α-Lipoic acid (ALA) is synthesized *de novo* in mitochondria from an 8-carbon fatty (octanoic) acid and functions as a cofactor for several mitochondrial complexes involved in the tricarboxylic acid cycle (TCA) cycle [116]. However, due to sterical differences only R-lipoic acid is endogenously synthesized and covalently bound to the acyl-carrier protein (ACP) in each multienzyme complex [117]. Such a non-protein cofactor known as “prosthetic group” activates the glycine cleavage system and four α-ketoacid dehydrogenase complexes, including: the pyruvate dehydrogenase, the α-ketoglutarate dehydrogenase, the branched-chain α-ketoacid dehydrogenase, and the 2-oxoadipate dehydrogenase complex.

ALA is a physiological compound produced in mammalian cells as part of their basic metabolism that can be safely administered to humans. In some countries ALA is considered a dietary supplement and in others a pharmaceutical drug. ALA is metabolized in different ways when given as a dietary supplement in mammals in various combinations; degradation of ALA is similar in humans and in rats [118]. The safety of ALA has been demonstrated in multiple clinical studies, including SYDNEY, SYDNEY 2, ALADIN I, II, and III, and NATHAN I and II [34,35,36,37,38,39,40,119].

One report of acute ALA-induced toxicity [120] was related to a suicidal attempt following ingestion of 18 g ALA that was, however, reversed after a 3-d supportive treatment. On the other hand, a body of literature has assessed the protective action of ALA against a number of xenobiotics in in vivo [121,122] and in vitro [102,103] investigations.

### 4.2. CoQ10

Coenzyme Q10 (coQ10) is a natural—and indispensable—compound present in mitochondria.

The synthesis of coQ10 starts with either tyrosine or phenylalanine to synthesize 4-hydroxybenzoate and the benzoquinone followed by the polyisoprenoid side chain synthesis from acetyl-coenzyme A (CoA) via the mevalonate pathway, and by the condensation of these two structures [123]. The benzoquinone group of coQ10 is critical for its function in OXPHOS and ATP production. Accepting electrons from reducing equivalents helps generate fatty acids and glucose metabolism, while transferring protons from the mitochondrial matrix to the intermembrane space creates a proton gradient across the inner mitochondrial membrane [123]. This feature also helps coQ10 to transport protons across lysosomal membranes to maintain the optimal pH [124].

The use of coQ10 as a dietary supplement offers very low toxicity and does not induce serious adverse effects in humans [125]. The acceptable daily intake is 12 mg/kg/day, calculated from the no-observed-adverse-effect level (NOAEL) of 1200 mg/kg/day derived from a 52-week chronic toxicity study in rats, i.e., 720 mg/day for a person weighing 60 kg according to Hidaka et al. [126]. coQ10 was well tolerated at up to 900 mg/day according to Ikematsu et al. [127]. In addition, administration of exogenous coQ10 does not inhibit the physiological production of coQ10 [128,129]. A recent study by Galeshkalami et al. [130] reported on the benefits of ALA and coQ10 combination on experimental diabetic neuropathy by modulating OS and apoptosis.

### 4.3. CARN

The amino acid derivative carnitine (CARN) is primarily synthesized in the liver in its L-form from lysine and methionine and transported via the bloodstream to cardiac and skeletal muscle [131]. It is required for mitochondrial fatty acid β-oxidation and transport of long-chain fatty acids across the inner membrane of the mitochondria, in the form of acyl-carnitine, where they can be metabolized for energy.

CARN and its active stereoisomer acetyl-L-carnitine (ALC) have been used in a number of human studies alone or as part of a combination therapy since early reports [132]. Administration of CARN in clinical studies including Alzheimer’s disease, depression, aging, diabetes, ischemia, and other neurological diseases did not report major toxic effects (reviewed in [7]). CARN/ALC and different chemical analogs have been used extensively as adjuvant treatment in neurological disorders or to prevent adverse side effects of different chemotherapeutic treatments, such as reducing brain injury after hypoxic-ischemia in newborn rats [133]. Song et al. [134] performed a metanalysis of randomized controlled trials and reported that CARN had good tolerance in patients with chronic heart failure improving clinical symptoms and cardiac functions.

Altogether, we can conclude that separate administration of ALA, or coQ10, or CARN is safe in human and in animal health. Although there are no reports of the combined use of the three MNs in humans, any combined administration should not present potential problems when administered in patients suffering from FA or T2D.

## 5. Clinical Determination of Mitochondrial Dysfunction

Traditionally any clinical evaluation of mitochondrial function in patients, especially for the diagnosis of neuromuscular diseases, has been performed using skeletal muscle biopsies due to the high number of mitochondria in this tissue and the relatively easy way to collect samples with very few side effects. However, due to the impaired hemostasis in FA patients, the use of clinical muscle biopsies is not advisable. Thus, it is recommended to use only non-invasive methods. A consensus statement of the Mitochondrial Medicine Society on diagnosis of mitochondrial diseases established some diagnostic methods that included the following measurements: (a) lactate and pyruvate levels in plasma; (b) plasma acylcarnitines, (c) urine organic acids, and (d) coQ10 levels in white blood cells (WBC) [135].

It would be advisable to study MDF and OS prevention by MNs in FA patients using biomarkers of mitochondrial function in plasma or blood cells from FA patients. We recommend—as it has been performed in other pathologies—the determination in white blood cells of mitochondrial OS-related enzymes like SOD2, the determination of the mitochondrial membrane potential from isolated white blood cells [136], or 8 hydroxy-deoxyguanosine as a marker of mitochondrial oxidative DNA damage [137]. It would be also advisable to determine well known markers of systemic OS in plasma samples as an indirect marker of mitochondrial impairment, such as the determination of the lipoperoxide biomarker malondialdehyde or calculation of the reduced/oxidized glutathione ratio, the most important antioxidant in mammalian cells (GSH/GSSG) in whole blood of patients [105,111].

## 6. Conclusions

As predicted by Luft in 1994 [1], “mitochondrial medicine” has met a broad and growing flow of basic and clinical investigations, with extensive implications in a number of disorders related to several medical disciplines [2,7]. This is the case for both T2D and FA, through different pathogenetic mechanisms and background information, and pointing to the prospects of utilizing mitochondrial-targeted adjuvant treatments. Thus, clinical studies are warranted in the improvement of life quality and expectation of patients affected by T2D and FA.

Highlights:♦T2D shows excess prevalence in patients with Fanconi anemia;♦Both T2D and FA display in vitro and in vivo oxidative stress;♦Mitochondrial dysfunction is recognized in T2D patients;♦Successful adjuvant treatments in T2D patients used one mitochondrial nutrient;♦The use of MN mixtures in treating T2D patients is suggested as a working hypothesis;♦MDF in FA was confined to molecular/in vitro studies as in vivo data are unavailable;♦Research into in vivo MDF in FA patients is warranted.

## Figures and Tables

**Table 1 antioxidants-09-00082-t001:** Reported clinical trials testing adjuvant administration of mitochondrial nutrients (MNs) in Type 1 and Type 2 diabetic patients (reviewed in [7]).

Mitochondrial Nutrients	No. Studies [Controlled Studies]	No. Treated Patients	Success Ratio
α-Lipoic acid	42 [30]	2980	0.93
Coenzyme Q10	9 [7]	370	0.89
(Acyl-)Carnitine	13 [9]	1894	1.00

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
