# Peer review of "Mitoprotective Clinical Strategies in Type 2 Diabetes and Fanconi Anemia Patients: Suggestions for Clinical Management of Mitochondrial Dysfunction"

_antioxidants, 2020, doi:10.3390/antiox9010082_

Round 1

Reviewer 1 Report

The present review is aimed at focusing on two apparently distant diseases, namely type 2 diabetes (T2D) and a rare genetic disease, Fanconi anemia (FA). The review explain well the relationship between the type 2 diabetes and Fanconi Anemia. It is written properly in a professional and understandable English language. It could be accepted in the present form.

Author Response

Also on behalf of co-authors, our sincerest thanks to Reviewer #1 for her/his appreciation of our work.

Reviewer 2 Report

In this Review by Pagano, et al., a case is made for assessing the status of oxidative stress and mitochondrial dysfunction in Fanconi anemia patients with an eye towards using mitochondrial nutrients as a clinical intervention for FA patients. This review is well-written and well-cited, and brings forward an appreciably relevant topic for both the fields of mitochondrial biology and FA studies. Therefore, in principle, the specific topic and the overarching goals of this Review merit publication. However, as detailed below, the authors should address a few items before this Review is fit for publication. To summarize this Reviewer's major critique, the body of this Review does not really give any additional information than is contained within the abstract. The body of the review simply restates each statement in the abstract in a different manner and cites each statement. A common theme in this paper is the authors state that a study was conducted or that a molecule is involved in MDF or OS, yet no explanation or details are provided as to what the results of the studies were or how a molecule is involved in MDF or OS. These simple statements are not sufficient for the stated scope of this journal. Readers are left with no real details related to the authors' goals.

The authors' should verify that the title really matches the goal of the paper.  An example of statements that should be supported by additional description is the sentence in lines 85-87. Please at least add a phrase to this sentence that summarizes the referenced studies rather than just stating that studies were performed. Another example is section 3.1 (lines 106-113). Here, readers are informed that T2D is a well-established phenotypic feature of FA. This is re-stating information provided in the Abstract and Introduction, but no additional information is provided in this short paragraph, which is one of the key sections of the Review justifying the authors' proposed clinical strategies. The bare statement of T2D association with FA is insufficient groundwork for the subsequent proposals in this review. Please spend a sentence or two describing the details. What is the prevalence? Does T2D stratify with severity of disease? Are there specific features of T2D that specifically accompany FA? Similarly, in section 3.2, the authors should at least provide brief details as to the nature of OS impinging on gene products, gene transcripts, cultured cells, etc. Because the authors are using these details to bolster the ultimate point of this review (clinical interventions designed to target these details), including these details is not outside the scope of this review, and in fact is critical to this review. The authors should provide more robust justification for the last statement of section 3.2? They previously cited Bagby's statement as the "extensive consensus of the scientific community". The statement here seems to moderately contradict Bagby's view (Bagby said OS is a non-canonical feature of FA, while the authors state that OS is a major pathogenetic feature of this disorder [also, did the authors mean "pathogenic"?]), so a brief justification would be helpful. Addressing the previous comment (#4 above) would also help justify their statement here. Again, in section 3.3 lines 121-123, the authors should briefly state how FA-G and perhaps other FA proteins are associated with MDF, instead of simply stating that they are associated with FA. For sections 4.1, 4.2, and 4.3, the authors really must provide brief explanations as to the nature of ALA, CoQ10, and CARN. At the minimum, because these molecules are a primary focus of this review, at least the biochemical and cellular roles of these must be explained.  There are some odd fonts in section 4.2.

It should be noted, that while this Reviewer has recommended several additions to this Review, these edits are not extensive. The authors have clearly already done the work of finding the citations and reading the literature related to these comments. They simply have to relate the actual information from these citations. Following these addition, the authors really have a nice Review article.

Author Response

The critical notes of Reviewer #2 were a precious ground for improving our manuscript. This underwent multiple changes according to the Reviewer’s suggestions, especially regarding the expansion of text in the sections that were highlighted. Moreover, English style and spelling were amended as thoroughly as possible.

Altogether, thanks are due to this Reviewer for her/his precious inputs leading to an improved manuscript.

Reviewer’s comments

An example of statements that should be supported by additional description is the sentence in lines 85-87. Please at least add a phrase to this sentence that summarizes the referenced studies rather than just stating that studies were performed. Another example is section 3.1 (). Here, readers are informed that T2D is a well-established phenotypic feature of FA. This is re-stating information provided in the Abstract and Introduction, but no additional information is provided in this short paragraph, which is one of the key sections of the Review justifying the authors' proposed clinical strategies. The bare statement of T2D association with FA is insufficient groundwork for the subsequent proposals in this review. Please spend a sentence or two describing the details. What is the prevalence? Does T2D stratify with severity of disease? Are there specific features of T2D that specifically accompany FA? Similarly, in section 3.2, the authors should at least provide brief details as to the nature of OS impinging on gene products, gene transcripts, cultured cells, etc. Because the authors are using these details to bolster the ultimate point of this review (clinical interventions designed to target these details), including these details is not outside the scope of this review, and in fact is critical to this review. The authors should provide more robust justification for the last statement of section 3.2?

Original submission text:

Lines 85-87… At the purpose to counteract T2D-associated OS, the use of several antioxidants was suggested [22-30]. Thereafter, and up to recent studies, MDF was further investigated as a salient feature of T2D,

Lines 106-113… MDF and OS in experimental diabetes or in other MDF-related disorders [67-73]…. Based on the available knowledge of successful use of each NM in T2D, and in several other disorders, one may suggest that safe administration of triple NM combinations might provide successful outcomes in counteracting T2D-related symptoms and progression. This working hypothesis warrants design of ad hoc experimental and clinical studies.

Section 3.2… Excess T2D prevalence

An excess T2D prevalence is a well-established phenotypic feature in FA patients [75-78], which is likely to underlie shared pathogenetic mechanisms, first including the role(s) of OS both in T2D and in FA. This clinical feature, of excess T2D prevalence, is not confined to FA, as far as a number of other genetic diseases [79-83] display excess T2D prevalence (along with OS features), independently of their multiple and different genetic deficiencies. Thus, one may speculate that the co-occurrence of these disorders and T2D is not coincidental, but suggest shared pathogenetic mechanisms focused on OS/MDF abnormalities, suggesting prospect study design toward clinical interventions in patients’ management.

Revised text:

Neverthess, it should be mentioned that MN combinations were reported in either in vitro or in animal studies aimed at counteracting MDF and OS in experimental diabetes or in other MDF-related disorders [67-73]. This was the case, e.g., for the studies by Shen et al. [69,70] reporting on combined ALA and CARN mixture-induced protective effects in rat or murine cells; MN mixtures were also found to protect diabetic rats from immuno- and liver dysfunction [72,73].

An excess T2D prevalence is a well-established phenotypic feature in FA patients [75-77], as assessed by the International Fanconi Anemia Registry [75], along with excess prevalence (81%) of multiple endocrine abnormalities. This finding was confirmed by further studies [76,77], both in FA patients and in FA mice [78]. These links between FA and T2D are consistent with shared pathogenetic mechanisms, first including the role(s) of OS both in T2D and in FA. It is worth noting that excess T2D prevalence is not confined to FA, as far as a number of other genetic diseases [79-83] display excess T2D prevalence (along with OS features), independently of their multiple and different genetic deficiencies. Thus, the co-occurrence of T2D in these disorders suggests  common deficiency(ies) focused on OS/MDF abnormalities

The occurrence of redox abnormalities in FA is recognized since early in vitro studies by Nordenson [84] and by Joenje et al. [85]. Subsequently, an extensive body of literature has assessed evidence for the pathogenetic roles of OS in FA gene products [86-94], gene transcripts [95], cultured cells [96-103], and in freshly drawn blood cells, plasma and urine from FA patients [104-106]. In brief OS, as a “non-canonical” feature of FA [74], can be reasonably considered as a major pathogenetic feature of this disorder.

Conclusion

I feel that the major points highlighted by Reviewer #2, i.e. providing details about the major references to be properly cited in text, were accomplished in the revised text, which now provides more detailed links with essential citations.

Regards,

Giovanni Pagano